# OpenReview forum: "SpecEval: Evaluating Model Adherence to Behavior Specifications"
_TMLR — Accepted by TMLR_

### Review · Reviewer_2K4N · 2026-02-08

**Summary Of Contributions:**

This paper introduces SpecEval, an automated framework for auditing whether foundation models adhere to their developers’ own published behavioral specifications. The core contribution is the notion of three-way consistency, which measures alignment between (i) a provider’s natural-language specification, (ii) the model’s generated outputs, and (iii) adherence judgments produced by a model-as-judge. To operationalize this audit, the authors propose a scalable pipeline that parses specification statements, adaptively generates targeted test prompts, and evaluates model responses using LLM judges guided by the specification. The framework is applied to 16 models from six major providers across more than 100 behavioral statements, revealing substantial consistency gaps between stated guidelines and observed behavior. The paper further includes human validation studies to assess both the quality of generated prompts and the reliability of LLM judges.

Some weakness:
- Specification ambiguity and conflict & Lack of formal semantics or guarantees
- Circularity in evaluation: Using provider models as both generators and judges risks reinforcing shared biases and conflating internal consistency with true adherence.
Limited human validation scale

**Audience:**

Yes

**Audience Explanation:**

- Relevance to evaluation and measurement research
- Connection to transparency, auditing, and accountability

**Broader Impact Concerns:**

The paper addresses an evaluation and auditing methodology rather than proposing new model capabilities, and therefore poses limited direct risk.

**Claims And Evidence:**

Yes

**Claims Explanation:**

The paper defines three-way consistency (provider spec ↔ provider model outputs ↔ provider model-as-judge scores) and reports provider-level averages showing notable gaps (e.g., Anthropic higher than OpenAI, Google lower), with differences on the order of ~20 percentage points between top and bottom providers.
They report that averaging across three different judge models yields scores that “closely track” the provider-specific three-way results (within a few points), which supports the claim of stability to judge choice.

For stronger interpretations like objective compliance, true adherence, or correctness, the evidence is not sufficient (and the method is not designed to provide that), so claims should be phrased carefully as judge-based adherence under natural-language rubrics rather than verified compliance.

**Requested Changes:**

The paper should more explicitly and consistently frame its results as judge-based adherence under natural-language specifications, rather than as objective or normative compliance. Several sections (especially qualitative analyses) could be misread as making stronger correctness claims than the methodology supports.

While the paper acknowledges some limitations, it should more explicitly discuss the implications of using provider models as both Candidates and Judges in the three-way consistency setting. A clearer explanation of what three-way consistency does and does not measure would help prevent overinterpretation of the results.

The paper should distinguish more clearly between (i) genuine behavioral violations by models and (ii) apparent violations caused by specification ambiguity, underspecification, or conflicting statements.

Without clarification, readers may incorrectly attribute all low scores to model failures rather than to limitations of the specifications themselves.

---

> ### Author Response · Authors · 2026-03-09
> **Response to Reviewer 2K4N**
>
> We thank the reviewer for their time and feedback. We address their concerns below.
>
> ## RE: Framing results as judge-based adherence under natural-language specifications
>
> We agree it is important to frame our results consistently. More specifically, we agree that our method should not be read as establishing objective compliance, formal correctness, or any kind of semantic guarantee. Our framework operates over natural-language specifications and natural-language judge rubrics, so **the quantity we measure is necessarily judge-based adherence under those rubrics rather than verified ground-truth compliance.**
>
> In the revision we therefore made the framing more explicit throughout the manuscript. We now clarify in the abstract, introduction, methods section, and qualitative analysis that three-way consistency is a judge-based measure of adherence under provider-written specifications. We also added a dedicated ``Scope of Three-Way Consistency'' paragraph in the methods section explaining what three-way consistency does and does not measure, so readers do not mistake it for a claim about objective behavioral correctness.
>
> ## RE: Circularity in using provider models as both Candidates and Judges
>
> We also agree it is important to be explicit about the implications of using provider models as both Candidates and Judges in the three-way setting. More precisely, in our setup the provider model is used as the Candidate and the Judge in the three-way slice, while the TestMaker is fixed separately. This design is deliberate: the goal of three-way consistency is not to provide an external verdict, but rather to measure whether a provider's own specification, model outputs, and model-as-judge are internally coherent with one another.
>
> For this reason, we view three-way consistency as a minimal internal consistency check. If a provider's own model, when used as judge, does not rate that provider's own outputs as adherent to that provider's own specification, then there is already a meaningful inconsistency inside the provider-controlled ecosystem. At the same time, we agree this does not remove concerns about circularity or self-favoring bias. In the revision we address this in two ways. First, we added explicit scope language in the new ``Scope of Three-Way Consistency'' paragraph explaining that low three-way consistency implicates the triple of specification, model, and judge rather than the model alone. Second, we point to the existing empirical checks that bound the concern. **The paper does not rely only on the same-provider Candidate/Judge slice: we also evaluate the same candidate models under judges from other providers and across other providers' specifications in the cross-judge and cross-spec analyses.** Those judge-averaged results closely track the provider-specific three-way results, which suggests that the main pattern is not being driven solely by the self-judge setting. We also include human validation showing that LM judges broadly track human labels rather than behaving as arbitrary self-confirming evaluators.

---

> ### Author Response · Authors · 2026-03-09
> **Response to Reviewer 2k4N (part 2)**
>
> ## RE: Distinguishing model failures from specification ambiguity or conflict
>
> We agree this distinction is important. A low score can arise either because a model genuinely fails to follow an unambiguous requirement, or because the specification itself is ambiguous, underspecified, conflicting, or overly rigid in a particular scenario. Without making this distinction explicit, readers could over-attribute low scores to model failures.
>
> In the revision we therefore strengthened the qualitative analysis to make this interpretive issue more explicit. More precisely, **we do not claim that our framework can cleanly identify whether any given low score is caused by model failure or by ambiguity in the specification.** That attribution often requires additional human judgment. What we can do is surface cases where the pattern of results makes specification-induced artifacts plausible, and contrast those with cases where the specification is comparatively clear and the models appear to behave differently under it.
>
> We revised the qualitative discussion with that narrower goal in mind. Rather than presenting a definitive decomposition, we now use concrete examples to illustrate two common interpretations of low scores: cases that appear more likely to reflect ambiguity, underspecification, or tension in the specification itself, and cases that appear more likely to reflect genuine behavioral differences between models. In the revised main paper, we make this more explicit with two representative examples: an imminent-harm / CPR case that plausibly reflects a specification-induced artifact, and a refusal-style case that more plausibly reflects a genuine behavioral difference under a comparatively clear rubric. We also clarify in the discussion that three-way inconsistency is still informative even when the source of the problem is not uniquely identifiable, because the goal of the audit is to assess the coherence of the full ecosystem rather than to assign blame to the model alone. Relatedly, we now make the interpretation of low scores more symmetric in the main results: a low three-way consistency score means that something in the triple of specification, model, and judge is inconsistent. The appropriate response may be to improve the model, revise the specification, or both.
>
> ## RE: Formal semantics, guarantees, and human validation
>
> We also agree with the broader concern that this framework does not provide formal semantics or guarantees. At the same time, we want to emphasize that we did not rely only on model-based evidence. **We conducted a comprehensive human study (Appendix F) with Prolific**, a platform for human evaluation and data labeling that is widely used and has been reported in prior work to yield higher-quality participants than MTurk:
>
> - https://journals.plos.org/plosone/article?id=10.1371/journal.pone.0279720
> - https://www.nature.com/articles/s41598-023-46048-5
>
> We recruited experienced annotators from Prolific's AI Tasker program with prior experience reviewing AI-generated outputs, fact-checking, and providing detailed feedback on AI systems. We focused on the OpenAI model spec, subsampled 20 statements, and for each statement selected one question and the corresponding responses from all three models, yielding $n = 60$ items to rate. In addition to this study, we also ran a separate human validation study on prompt relevance with three experiences annotators.
>
> Our point is therefore not that the paper provides exhaustive guarantees over every possible prompt-response pair. Doing that exhaustively over the full prompt set would be prohibitively expensive. Rather, we believe the human studies we did run are appropriately targeted and have enough power to support the main methodological claims of the paper: that the generated prompts are relevant to their intended statements, and that LM-as-judge behavior tracks human annotators reasonably well on a meaningful sample.

---

> > ### Author Response · Authors · 2026-03-09
> > **Response Part 3**
> >
> > To summarize, we have made the following revisions:
> >
> > 1. Clarified throughout the paper that our results should be interpreted as judge-based adherence under natural-language specifications rather than objective or normative compliance.
> > 2. Added a dedicated ``Scope of Three-Way Consistency'' paragraph in the methods section explaining what three-way consistency does and does not measure.
> > 3. Clarified that, in the three-way setting, the provider model is used as Candidate and Judge as part of a deliberate internal-consistency design, while the TestMaker is fixed separately.
> > 4. Added and retained language explaining that low three-way consistency implicates the provider ecosystem (specification, model, and judge) rather than the model alone.
> > 5. Made clearer that the paper also includes cross-judge and cross-spec analyses, so the same candidate models are compared under non-native judges and non-native specifications rather than only in the self-judge setting.
> > 6. Strengthened the qualitative analysis to discuss likely specification-induced artifacts alongside likely genuine behavioral differences, while making clear that this is an interpretive analysis rather than a formal decomposition.
> > 7. Connected the main results more clearly to the cross-spec analysis so that readers can see how specification-level difficulty contributes to low scores.
> > 8. Added and retained a dedicated Limitations section discussing model-as-judge limits, specification ambiguity, and the broader interpretive limits of the framework.

---

### Review · Reviewer_LqRR · 2026-02-23

**Summary Of Contributions:**

The paper „SpecEval: Evaluating Model Adherence to Behavior Specifications” presents a study that assess how well LLMs adhere to behavioral guidelines from three major model providers, i.e., OpenAI, Anthropic, and Google. The core rationale is that models should at least be behaviorally compliant, when they judge themselves according to their own guidelines (called three-way consistency). The results show that models from OpenAI and Anthropic adhere closer to their specs and that this is also the case when other models are used as a judge.

Strengths:

-	Interesting benchmarking strategy. I especially like the self-consistency approach that demonstrates that the models themselves are currently sufficient to show how they are non-compliant.
-	Consideration of development over time and of open-weights models
-	LLM-judgments compared with human judgments.

Weaknesses:

-	Behavioral specs are not compared to each other, leading to possible issues when just comparing adherence accuracy
-	There is a taxonomy for behavioral specs, but I cannot see why it is introduced or how it is used at all for evaluation purposes.
-	The results of the open weights models are basically ignored.
-	Possible bias by only using a single model (GPT-4.1) to generate the tests.
-	No discussion of limitations.
-	Poor formatting that even leads to information not being visible in the Appendix.

**Audience:**

Yes

**Audience Explanation:**

Behavioral compliance is an important topic for the practical application of language models. The benchmark adds a new perspective.

**Claims And Evidence:**

No

**Claims Explanation:**

1) My key concern is that the three behavioral specs are not compared to each other. Even a naïve consideration of the number of statements per spec shows that while OpenAI (46) and Anthropic (49) are fairly similar, the Sparrow spec from Google (23) contains only about half as many statements. This could mean different things that lead to very different interpretations of the results of, e.g., Figure 4 with the main results for the three-way consistency. Here are two example scenarios for how the interpretation of the results could be affected, based on different relationships between the specs:
Scenario 1: Sparrow could have more complex statements that mix concerns that are separated in the OpenAI and Anthropic specs. If the Sparrow-spec would cover the same behavioral principles as the OpenAI and Anthropic specs, this would mean that each statement in the Sparrow-spec would, on average, cover twice as many concerns. If one concern is missed, the whole statement would not be fulfilled, leading to a more severe drop in accuracy as when compared to the OpenAI and Anthropic specs, where one statement might be fulfilled on one not. Overall, this could mean that the compliance would be similar to that of OpenAI and Anthropic, when this would be considered.
Scenario 2: Sparrow could just have less statements than OpenAI and Anthropic. In this case, the Google models would have the lowest compliance, even though there are less rules.
As you can see, the two (extreme) scenarios lead to vastly different interpretations of the results. Therefore, I strongly believe this needs to be considered to understand possible biases in consistency scoring.

2) The paper introduces quite a few things that are never explored, including a taxonomy of the behavioral specs and the applying of proprietary specs to open-weights models, but also the qualitative results in Section 4.4. This needs to be cleaned up or properly used/discussed within the paper.

3) The test data is currently only generated with GPT-4.1. While the human judgments show that the data seems to be good, it is still unclear whether this introduces biases in favor or against GPT-4.1. A possible source for biases – that the validation did not and cannot reasonably consider – is that the style of the generated tests might be GPT-4.1 specific. Such a style-adherence could be positive for GPT-4.1, e.g., if this matches patterns for instruction-following that OpenAI generated during their own training of the models *helping* the model to avoid the issues. A less likely case is that these are exactly the cases that mimic GPT-4.1 weaknesses, making the conformance especially difficult for GPT-4.1. In any case, a comparison with a second prompt-generator would solve these possible biases.

4) There is no discussion of limitations. Unless there are none, this needs to be added.

5) The formatting is partially poor and needs to be fixed. Many tables in the Appendix run over the end of the pages, often leading to missing information. There are also subsubsections without subsections in the appendix, leading to broken section numbering.

**Requested Changes:**

- Compare specifications with each other and use this to understand how differences affect the behavioral adherence measures for specific specs.
- Consider using a second test generator or explain why there is no bias for the GPT-4.1 results.
- Cleanup or properly discuss all content.
- Fix the formatting of the paper.

---

> ### Author Response · Authors · 2026-03-09
> **Response to Reviewer LqRR**
>
> We thank the reviewer for their time and feedback. We address their concerns below.
>
> ## RE: Behavioral specs not being compared to one another
>
> We agree it is important to consider how specifications compare with one another, though we want to emphasize that the **three-way consistency score in Figure 4 is intentionally not the cross-spec comparison view**. Our motivation is that each provider publishes its own specification and also controls the model and judge used in the three-way setting, so we wanted one view of the data that keeps those artifacts together as much as possible rather than immediately imposing a shared external comparison.
>
> For this reason, we do not claim that three-way consistency captures complete behavioral soundness or that it should be read as a clean cross-provider ranking. Instead, it is meant to capture a minimal internal consistency question: can a provider's model, when evaluated by that provider's own judge model, satisfy that provider's own published specification? We view this as a deliberately charitable baseline. If a model fails even this within-provider test, then there is already a meaningful inconsistency between artifacts the provider directly controls.
>
> At the same time, we agree with the reviewer's underlying concern that differences between specifications can confound cross-provider interpretation. This is exactly why we include Figure 5. Concretely, **Figure 5 is meant to control for this confound by evaluating the same set of candidate models and averaging over the same set of judges across all three specifications.** Under that view, scores are lower on Sparrow for all models, not only Google models, which indicates a specification-level difficulty effect in addition to model-level effects. We also report scores on a 0-1 scale, which means the number of statements does not by itself mechanically inflate or deflate the aggregate score for a specification.
>
> We agree our presentation could be clearer, so in the revision we have:
> - explicitly stated near Figure 4 that these are within-provider scores and should not be interpreted as a standalone cross-provider leaderboard;
> - revised the following results subsection to foreground Figure 5 as the cross-spec comparison view.
>
> ## RE: Taxonomy and open-weight models
>
> We agree the taxonomy is not critical to the evaluated contribution in this paper. We originally included it as an initial attempt to organize the space of behavioral specifications we observed across providers, in the hope that future work could build on it. However, because we do not actually use that taxonomy in the evaluation pipeline or in the interpretation of the main results, we agree that it creates unnecessary scope in the current paper. For that reason, we removed it from the appendix and kept only a brief future-work note that a more unified taxonomy of model-provider specifications would be valuable.
>
> For the open-weight models, we also agree that the original paper did not provide enough interpretation. These providers do not publish comparable behavioral specifications, so we cannot compute three-way consistency for them and we do not want to imply that their cross-spec scores should be read as compliance judgments. Our intention in including them was more limited: they provide a point of reference for whether there appear to be shared norms across models that were post-trained with similar kinds of data and alignment procedures. We now make that framing explicit in the results section and state more clearly that **these are behavioral reference snapshots under external specifications, not judgments against their own declared policies.**
>
> We also agree that the qualitative analysis section needed to be better tied back to the paper's main argument. In the revision we clarify its role by using representative examples to illustrate cases that plausibly reflect specification-induced artifacts and cases that plausibly reflect genuine behavioral differences between models, while making clear that this is an interpretive analysis rather than a formal decomposition.

---

> > ### Author Response · Authors · 2026-03-09
> > **Response to Reviewer LqRR (Part 2)**
> >
> > ## RE: GPT-4.1 as a single TestMaker
> >
> > We appreciate the concern about potential TestMaker bias. More generally, we agree that using a single model to generate evaluation prompts is a real methodological limitation and deserves explicit discussion.
> >
> > At the same time, several aspects of the results make a strong GPT-4.1-specific advantage less likely. **Empirically, GPT-4.1 does not receive the highest adherence outcomes in our main analyses, including the cross-spec comparisons in Figure 5.** If the generated prompts strongly favored GPT-4.1, we would expect GPT-4.1 to dominate those comparisons, but that is not what we observe. In addition, the main comparative results are averaged across three judges from different providers, which reduces dependence on any single model family's preferences. We also report human-validation agreement between model judges and human raters, which gives us some evidence that the signal is not merely GPT-4.1-specific style matching.
> >
> > That said, we agree this does not remove the limitation. Using a single TestMaker remains a limitation of our work, and we now state that explicitly in the Limitations section. We also agree that using a second TestMaker would be a valuable robustness check. Practically, re-running the full pipeline would require substantial additional API cost, repeated validation work, and some of the originally evaluated models are now deprecated, but we view that as a reason to scope this as future work rather than dismiss the concern.
> >
> > ## RE: Limitations
> >
> > We agree it is important to make the limitations of our work explicit. We had originally tried to gesture at these issues through the conclusion and future-work discussion, but we agree that this was not sufficient. In the revision we therefore added a dedicated Limitations section that directly discusses the main constraints of the current study: the use of a single TestMaker, the restriction to single-turn interactions, the temporal snapshot nature of the audit, model-as-judge circularity, and cases where low scores may reflect ambiguity or limitations in the specification itself rather than only model failures.
> >
> > ## RE: Formatting
> >
> > We apologize for the formatting issues. We revised the appendix formatting and structure (including table rendering and section hierarchy) and did an additional compile pass to reduce clipping/overflow issues.
> >
> > To summarize, we have made the following revisions:
> >
> > 1. Clarified in the Results section (near Figure 4) that three-way consistency is a within-provider internal consistency metric and not a standalone cross-provider leaderboard.
> > 2. Revised the subsection following Figure 4 to foreground Figure 5 as the cross-spec comparison view (fixed candidate set, shared judges, cross-spec interpretation).
> > 3. Kept and emphasized the cross-spec control interpretation in Figure 5, including the observation that Sparrow is harder across models.
> > 4. Retained and clearly presented spec-adjusted comparisons alongside raw three-way scores.
> > 5. Removed taxonomy material from the appendix and kept only a brief future-work note on unified taxonomy development.
> > 6. Expanded interpretation of open-weight model results, explicitly framing them as behavioral reference snapshots rather than compliance judgments.
> > 7. Clarified and strengthened Section 4.4 qualitative analysis, explicitly distinguishing specification-induced artifacts from genuine behavioral differences.
> > 8. Added and retained a dedicated Limitations section covering single TestMaker, single-turn scope, temporal snapshot effects, model-as-judge circularity, and specification ambiguity/quality limits.
> > 9. Explicitly acknowledged single-TestMaker generation as a limitation and noted multi-TestMaker robustness checks as future work.
> > 10. Finalized appendix formatting fixes (table sizing/layout, section hierarchy, and final compile pass to check clipping/overflow).

---

> > > ### Comment · Reviewer_LqRR · 2026-03-09
> > >
> > > Thank you for the response and the clarifying changes.
> > >
> > > Regarding the specification difficulty, I am not convinced. The analysis via Figure 5 is extremely indirect and possibly noisy due to the judges, does not address my concern, and demonstrates the opposite of what is claimed in the paper.
> > >
> > > The authors write: “When the specification is held constant, the relative ordering of provider models is preserved: on each of the three specifications, Claude-3.7-Sonnet scores highest, followed by GPT-4.1, then Gemini-2.0-Flash”.
> > >
> > > This is *not* what Figure 5 shows. Gemini is tied with GPT on the OpenAI Spec, and better than GPT on the Google and Anthropic specs. This *directly demonstrates* my concern instead: just using internal consistency (Figure 4) would mean “Gemini is least compliant”, whereas Figure 5 shows that this is an artifact of the specification difficulty and the compliance is actually the same or higher than for GPT.
> > >
> > > Thus, my concern remains: without a better consideration of the specification difficulty, the value of this is limited and a sound interpretation of results is hard. Indeed, if the metric would be adopted without such a consideration, it would add an incentive to use a simpler (public) specification, just to “beat the benchmark”.
> > >
> > > -------------------------------
> > >
> > > Regarding GPT-4.1 as the only source of data, listing that as limitation is okay for me, if done properly, including explaining *why this is a limitation and what the negative consequence could be*. Right now, the discussion of this limitation is restricted to mitigated factors. Based on the response as well, I believe this is due to faulty reasoning: The authors claim that because the scores are not better, there is likely not a bias. However, this assumes that “using the same model would lead to best scores”. Reality is “using the same model might increase scores” which means the limitation is that the scores are *overestimated* in comparison to the others, regardless of whether they are now best or now. That they are not best just means they are not grossly overestimated.
> > >
> > > Further, that the prompts are valid and judged by humans is besides the point. This is technical issue due to the linguistic style of data used for post-training, something that human judges do not consider. The causal chain is this: Premises: OpenAI likely uses GPT-generated prompts for the post-training. Alignment works best for data that is similar to post-training data. It follows that alignment for GPT-generated prompts likely works better for OpenAI model than for other data sources. From this follows that the values for self-assessment of GPT might be overestimated in comparison to the others.

---

> ### Author Response · Authors · 2026-03-11
> **Response to Reviewer LqRR**
>
> Thank you for the response. You are right that our revised text overclaimed what Figure 5 supports. In particular, the sentence stating that the relative ordering is preserved across all three specifications is incorrect, and we have removed it in a revised version uploaded now.
>
> We agree that Figure 5 is not a perfect control for specification difficulty and does not fully resolve the concern, since it still relies on LM judges and cannot fully control for differences in specification granularity or wording. However, we still think it is informative because it evaluates the same set of candidate models under the same three-judge averaging procedure across all three specifications. That makes the cross-spec comparison more standardized than Figure 4, even though it remains indirect and imperfect. We therefore use Figure 5 as partial evidence of a specification-level effect, not as a definitive normalization or benchmark.
>
> Our intended claim about three-way consistency is narrower. Three-way consistency is a within-provider internal-consistency audit: it asks whether a provider's own model, evaluated by that provider's own model-as-judge, is rated adherent to that provider's own published specification. A low score is informative even though it does not by itself identify whether the source is the model, the judge, the specification, or their interaction. We also agree with your incentive point. If three-way consistency were used as a benchmark, providers could be rewarded for publishing easier public specifications. That is not the use we advocate. Our use is provider-internal auditing against a provider's own declared standard, not benchmarking providers against one another.
>
> We also note that surfacing specification difficulty as a confound is itself a useful contribution. If providers use these documents to guide alignment — whether through constitutional AI approaches, RLHF reward modeling, or LM-as-judge evaluation — then understanding how specification wording and granularity affect measured adherence is directly relevant. Our results show this effect is real: the same model can appear more or less adherent depending on which specification it is evaluated against, which has practical implications for how specifications should be written and used.
>
> To make that scope explicit in the paper, we added a dedicated "Scope of Three-Way Consistency" paragraph that states what the metric does and does not measure. In particular, it explains that three-way consistency does not establish ground-truth correctness, does not show that the specification itself is adequate, and does not imply that an independent human panel would agree with every rating. It also makes explicit that low three-way consistency implicates the triple (specification, model, judge) rather than the model alone. We intended this paragraph to properly explain the limitations of three-way consistency and to prevent benchmark-style over-interpretation.
>
> We hope this clarifies both the limitation of the original claim and the narrower role that Figure 5 plays in our interpretation.
>
> Accordingly, we have revised the results section passage discussing Figure 5 in the updated manuscript as follows:
>
> ```tex
> To compare behavior across specifications, Figure~\ref{fig:llm_compliance_heatmap} provides a more standardized, though still imperfect, cross-spec view: it evaluates the same set of provider models on all three specifications and averages scores over the same three judges. Under this view, the relative ordering of GPT-4.1 and Gemini-2.0-Flash is not stable across specifications, illustrating that specification choice is a meaningful confound in cross-provider interpretation of Figure~\ref{fig:three_way_consistency}. All models also receive lower judge-averaged adherence on the Sparrow specification. We therefore interpret Figure~\ref{fig:llm_compliance_heatmap} as evidence that specification choice materially affects observed adherence scores, rather than as a definitive normalization that removes all cross-spec confounding. For this reason, Figure~\ref{fig:three_way_consistency} should not be read as a cross-provider leaderboard or benchmark. Instead, our primary claim for three-way consistency is narrower: it is a within-provider internal-consistency audit of the triple (specification, model, judge), while Figure~\ref{fig:llm_compliance_heatmap} provides contextual evidence about specification-level effects.
> ```

---

> > ### Author Response · Authors · 2026-03-11
> > **Response to Reviewer LqRR**
> >
> > ## RE: GPT-4.1 as the only source of data
> >
> > We appreciate this clarification and agree that our original discussion of the single-TestMaker limitation was insufficient. We had initially understood the concern as being about judge-side self-preference and therefore framed our mitigation around the fact that GPT-4.1 does not score highest. We now understand that your concern is on the TestMaker side, not the judge side. Because OpenAI likely uses GPT-generated text during post-training, their models may be better aligned to the linguistic style of GPT-generated prompts specifically. This means GPT-4.1's adherence scores could be inflated relative to what they would be under prompts from a different TestMaker, regardless of whether GPT-4.1 scores highest overall. We also agree that human validation of prompt relevance does not address this issue, since it checks whether prompts test the intended statements, not whether their linguistic style is closer to one model family's post-training data. The fact that GPT-4.1 does not score highest therefore only suggests that any inflation is not gross; it does not rule it out.
> >
> > We have revised the Single TestMaker limitation in the paper to explain the mechanism (stylistic similarity between GPT-generated test prompts and OpenAI's post-training data may advantage OpenAI models) and the consequence (GPT-4.1's scores may be overestimated relative to other models), rather than only listing mitigating factors.

---

### Review · Reviewer_w7fC · 2026-03-01

**Summary Of Contributions:**

Foundation model are often accompanied with published behavioral specifications (e.g., Anthropic’s “Constitution” and OpenAI’s “model spec”), but it is unclear how reliably models follow them in practice. This is important because model behavior and not just capability affects real-world usefulness and risk. There have been several examples such as April 2025 GPT-4o update that increased sycophancy despite stated guideline, highlighting the need for independent, third‑party auditing. This paper presents the first systematic audit of models against their own providers’ published behavioral specs.

**Audience:**

Yes

**Audience Explanation:**

Several outputs of the paper would be of interest to TMLR's audience:
- the automated testing pipeline,
- a large prompt dataset spanning multiple providers' specifications with responses and model‑judge annotations,
- the new three‑way consistency metric with statement‑level analyses (including cases where spec wording may be problematic), and
- human validation studies to check judge-label quality and whether generated prompts actually test the intended spec statements.

**Broader Impact Concerns:**

There are no broader impact concerns

**Claims And Evidence:**

Yes

**Claims Explanation:**

The paper introduces an automated evaluation pipeline that
(1) generates targeted test prompts for each specification statement using a “TestMaker” model
(2) collects answers from a “Candidate” model, and
(3) uses a “Judge” model to score whether the response adheres to the statement.
Their main metric is “three-way consistency,” which measures adherence when the provider’s own model is used both as Candidate and Judge against that provider’s specification, while also reporting cross‑model judging for broader ecosystem comparisons.
This is a rather simple but useful idea.

The paper audits 16 models from OpenAI, Anthropic, Google, Meta, DeepSeek, and Alibaba, and reports its findings:
1) Anthropic models are the most adherent overall (e.g., claude-3-7-sonnet averages 84% on Claude’s Constitution).
2) OpenAI is close behind but shows weaker adherence on certain important statements such as avoiding political manipulation and giving regulated advice, while tending to do better on avoiding judgmental refusals and presenting varied political opinions.
These findings could be useful observations for the community.

**Requested Changes:**

Some of the examples in qualitative analysis in the appendix can be pulled into the main text of the paper. This will help improve the paper's clarity.

---

> ### Author Response · Authors · 2026-03-09
> **Response to Reviewer w7fC**
>
> Thank you for your review and positive feedback. We appreciate your time and analysis.
>
> We are glad that you find the qualitative examples illuminating. We will revise our current section 4.4 to move the qualitative examples from the appendix into the main body to improve clarity of your work as suggested.

---

### Author Response · Authors · 2026-03-09
**Response to action editor and all reviewers**

We thank the action editor and reviewers for the discussion.  Across the discussion, we see three concerns. First, whether three-way consistency should be interpreted as objective compliance or instead as a more limited judge-based quantity. Second, whether differences across provider specifications materially affect cross-provider comparisons. Third, whether the candidate--judge self-loop and the use of a single TestMaker are decision-critical or instead bounded design choices. In our revision, we treated the second as the most important one, while also clarifying the first and bounding the third with framing and existing empirical evidence.

**We have uploaded a revised paper with all changes in Blue text that address the finer points for each reviewer and also the points below**

## RE: Interpretation of three-way consistency

We agree that three-way consistency should not be read as objective compliance. Because the framework operates over provider-written specifications and judge rubrics, the quantity we measure is judge-based adherence under those rubrics. The intended claim is narrower: three-way consistency asks whether a provider's specification, model outputs, and own model-as-judge are mutually coherent. If the provider's own judge does not rate the provider's outputs as adherent to the provider's own published specification, that is already meaningful evidence of an internal ecosystem inconsistency. We therefore view the main issue as one of scope rather than a deeper validity failure. In the revision, we made that scope explicit throughout the manuscript, including a dedicated ``Scope of Three-Way Consistency'' paragraph in the methods section and consistent framing in the abstract, introduction, results, qualitative analysis, and conclusion.

## RE: Differences across behavioral specifications

We agree this is the central issue. Raw cross-provider differences cannot be interpreted cleanly unless specification differences are controlled, which is why the revision separates Figures 4 and 5.

Figure 4 is now framed as a within-provider internal-consistency audit, not a standalone cross-provider leaderboard. Figure 5 is the controlled cross-spec view: it holds the candidate models and judge panel fixed while varying only the specification.

Under that controlled view, the same pattern remains. When the specification is held constant, the relative ordering of the main provider models is preserved across the Anthropic, OpenAI, and Sparrow specifications. At the same time, all models score lower on Sparrow, which shows a real specification-level effect in addition to a model-level effect. This directly addresses Reviewer LqRR's central concern: the results are not only a ranking of specification difficulty. Because each provider chooses its own standard, a provider whose model cannot reliably satisfy that standard still provides evidence of an ecosystem inconsistency. The revised paper therefore presents both views together: Figure 4 as the within-provider measure, and Figure 5 as the controlled cross-spec view.

Relatedly, the qualitative analysis now includes two main-text examples: one that plausibly reflects a specification-induced artifact, and one that more plausibly reflects a genuine behavioral difference under a clearer rubric. We present these as interpretive examples, not a formal decomposition.

## RE: Circular candidate--judge setup and single TestMaker

We agree this issue matters, but we do not think it is decision-critical for the main conclusions. In the three-way setting, the provider model is intentionally used as both Candidate and Judge because the goal is to test internal coherence, not to provide an external verdict. At the same time, the paper does not rely only on this same-provider slice: Figure 5 evaluates the same candidate models across all three specifications while averaging over all three LM judges, and these judge-averaged cross-spec results track the main three-way pattern closely. We also report cross-judge results, and human validation shows that LM-judge labels track human annotators. In the revision, we make this limitation more explicit by noting that the LM judges are black-box frontier models that may introduce uncertainty, bias, and misalignment with human judgment, even if those concerns appear reasonably bounded in our setting.

We also agree that using a single TestMaker is a limitation, and we now say so explicitly. However, the existing evidence suggests it is not a decision-critical flaw: GPT-4.1 does not dominate the main comparisons, Claude remains strongest overall, prompts were manually curated, and a separate human study shows that sampled prompts are relevant to their intended specification statements. We also agree that a more self-contained loop, where TestMaker, Candidate, and Judge all come from the same provider, would be useful follow-up work.

---

### Decision · Action_Editor_5MmC · 2026-04-07

**Recommendation:** Accept with minor revision

**Additional Comments:**

Suggestions for revision
- Scope and claim framing: clarify that the paper measures judge-based adherence rather than objective compliance, and that three-way consistency is a within-provider internal-consistency audit, not a cross-provider benchmark.
- Specification comparability: better address the confound of different specification difficulty / granularity / wording across providers, and be more careful in interpreting cross-spec comparisons.
- Methodological limitations: discuss more explicitly the risks from using a single TestMaker (GPT-4.1) and from candidate-judge circularity / self-loop evaluation.
- Result interpretation: distinguish more clearly between specification-induced artifacts and genuine model behavioral failures, and tie the qualitative analysis more directly to this distinction.
- Paper presentation: ensure the limitations section is concrete and complete, integrate qualitative examples cleanly into the main text, and fix remaining formatting / organization issues.

**Audience:**

Yes

**Audience Explanation:**

paper addresses a timely and practically important question in LLM evaluation, auditing, and transparency by providing a scalable way to assess whether models follow their developers’ own behavioral specifications.

**Claims And Evidence:**

Yes

**Claims Explanation:**

The paper provides clear and reasonably convincing evidence for its main, properly scoped claim: an automated framework can audit judge-assessed adherence of models to provider-written behavioral specifications, and this reveals substantial internal-consistency gaps. The empirical study is broad, the methodology is clearly described, and the inclusion of human validation materially strengthens the paper. That said, the evidence supports a narrower interpretation as judge-based adherence under natural-language rubrics, rather than objective compliance or a definitive cross-provider comparison.